# Cancer and Myotonic Dystrophy

**DOI:** 10.3390/jcm12051939

**Published:** 2023-03-01

**Authors:** Eleonora S. D’Ambrosio, Paloma Gonzalez-Perez

**Affiliations:** 1Department of Neurology, Nationwide Children’s Hospital, Columbus, OH 43205, USA; 2Department of Neurology, Massachusetts General Hospital, Boston, MA 02114, USA

**Keywords:** cancer, myotonic dystrophy, repeat expansion, tumor, malignancy screening

## Abstract

Myotonic dystrophy (DM) is the most common muscular dystrophy in adults. Dominantly inherited CTG and CCTG repeat expansions in *DMPK* and *CNBP* genes cause DM type 1 (DM1) and 2 (DM2), respectively. These genetic defects lead to the abnormal splicing of different mRNA transcripts, which are thought to be responsible for the multiorgan involvement of these diseases. In ours and others’ experience, cancer frequency in patients with DM appears to be higher than in the general population or non-DM muscular dystrophy cohorts. There are no specific guidelines regarding malignancy screening in these patients, and the general consensus is that they should undergo the same cancer screening as the general population. Here, we review the main studies that investigated cancer risk (and cancer type) in DM cohorts and those that researched potential molecular mechanisms accounting for DM carcinogenesis. We propose some evaluations to be considered as malignancy screening in patients with DM, and we discuss DM susceptibility to general anesthesia and sedatives, which are often needed for the management of cancer. This review underscores the importance of monitoring the adherence of patients with DM to malignancy screenings and the need to design studies that determine whether they would benefit from a more intensified cancer screening than the general population.

## 1. Introduction

Myotonic dystrophy (*dystrophia myotonica* or DM) is the most common muscular dystrophy in adults. It is an autosomal dominant multisystem disease with two different clinical forms: myotonic dystrophy type 1 (DM1, OMIM 160900) and myotonic dystrophy type 2 (DM2, OMIM 602668). The prevalence of DM1 varies widely across different populations, being higher in northern Sweden, Quebec, and the Basque region, suggesting a founder effect (prevalence = ~1/4000–8000) [1,2]. DM1 is caused by a CTG trinucleotide expansion (>50 repeats) within the 3′UTR of the *dystrophia myotonica protein kinase* (*DMPK)* gene on chromosome 19 [3]. The prevalence of DM2, although lower than that of DM1 in most studies, has been reported to be as high as 1 in 1830 in the Finnish population [4], which suggests that this disease form is probably underdiagnosed. DM2 is caused by a CCTG tetranucleotide expansion (>75 repeats) within intron 1 of the *cellular nucleic acid-binding protein* (*CNBP)* gene (formerly called *zinc finger protein 9*, or *ZNF9*) on chromosome 3 [5]. DM1 and DM2 share a common pathogenic mechanism; expanded DNA repeats are transcribed into expanded mRNA transcripts that accumulate within the nuclei of the cells forming the RNA foci. These mutant CUG-containing RNA strands sequester important splicing regulator proteins, such as CUGBP/Elav-like (CELF1) and muscleblind-like 1 (MBNL1). Thus, the functions of CELF1 and MBNL1 when these proteins are sequestered within these ribonuclear foci become impaired, resulting in the expression of an abnormally spliced variety of mRNAs that appear to be responsible for the multisystem involvement of these muscular dystrophies (Figure 1). Thus, the abnormal splicing of the chloride channel (*CLCN1*) is associated with myotonia, that of the insulin receptor (*INSR*) with insulin resistance, that of the calcium channel (*Cav1.1*) with muscle weakness, and that of the sodium channel (*SCN5A*) with arrhythmias [6,7,8,9,10].

Both the somatic instability of the repeat expansion across different tissues (somatic mosaicism) and the presence of epigenetic modifications appear to contribute to the variable type and severity of the clinical manifestations of DM [6,7]. Furthermore, the abnormal regulation of miRNAs in both DM1 tissue and extracellular vesicles in the biofluids of patients with DM1 have been postulated to play a key role in DM pathogenesis and even serve as disease biomarkers [11].

The molecular mechanism underlying carcinogenic events in DM is uncertain, but several hypotheses have been postulated, as discussed below.

### Myopathies and Cancer

In addition to myotonic dystrophies, several other myopathies have been associated with cancer or benign tumors. Thus, sporadic inclusion body myositis has been associated with T-cell granular lymphocytic leukemia/B-cell chronic lymphocytic leukemia; dermatomyositis has been associated with lung, ovary, breast, colorectal, cervical, bladder, nasopharyngeal, esophageal, pancreatic, colon, and kidney cancers; immune-mediated necrotizing myopathies have been associated with lung, ovary, breast, kidney, and gastrointestinal tract cancers; idiopathic myositis and myasthenia overlap syndrome have been associated with thymoma; late-onset sporadic nemaline myopathy has been associated with monoclonal gammopathy; and glycogen storage disease type 1a (von Gierke disease) has been associated with hepatocellular adenoma/adenocarcinoma [12].

Thus, inflammatory or immune-mediated myopathies, and myotonic dystrophies are the myopathies that are associated with a greater variety of organ-related cancers. Although dermatomyositis and immune-mediated necrotizing myopathies (both treatable myopathies) are associated with several types of cancers, there are specific antibodies that can be determined in the serum of these patients to stratify their cancer risk. For example, a patient with dermatomyositis and positive Tif-1 antibodies is at a higher risk of having an associated cancer, and intensive malignancy screening would be highly recommended in this scenario [13]. However, for myotonic dystrophies (for which no cure exists at this time), there are no stratifiers of cancer risk to guide malignancy screening in these patients. As we discuss later, cancer-related deaths in DM populations are not unsignificant, and whether malignancy screenings of specific cancers should be intensified in these patients is at this time unknown. 

## 2. Mortality in DM

We conducted a review of the literature using “myotonic dystrophy [Title/Abstract], cancer [Title/Abstract], mortality, and death [Title/Abstract]” in PubMed to identify research articles that studied death due to cancer in DM. The risk of premature death in DM is mostly due to the increased risk of respiratory (mostly pneumonia) and cardiac (mostly arrhythmias) complications, which are the first and second causes of death in patients with DM, respectively. Cancer has been reported to be the third most common cause of death in DM, accounting for up to 10–15% of deaths. Although mortality studies have been mostly carried out in DM1 cohorts, it is plausible that cancer-related death in patients with DM2 is greater than currently recognized because respiratory and cardiac complications in this patient subgroup are less frequent, and because this myotonic dystrophy is often underdiagnosed. Table 1 summarizes the studies that investigated cancer-related deaths in DM [14,15,16,17,18,19].

## 3. Carcinogenesis in DM

Although DM1 and DM2 appear to share the same pathogenic mechanism (Figure 1), the association between DM and cancer and its molecular basis has been mostly studied in DM1. We conducted a literature search in PubMed using “myotonic dystrophy [Title/Abstract] and cancer {Title/Abstract]” to identify research articles that investigated the frequency, type, and pathogenic mechanism of cancer in DM. The research articles that did not aim to investigate these aspects and non-research articles were also reviewed and referenced here only if considered clinically relevant for the purpose of this review. Of note, a few studies included patients with DM2 or undifferentiated DM in their DM1-predominant cohorts [20,21,22,23,24]. 

We differentiated two types of studies that link DM1 and carcinogenesis: one type includes those that investigated the cancer risk and predisposition to specific cancer types in predominantly DM1 cohorts by using the general population (or healthy subjects) or a non-muscular dystrophy cohort as a comparator group, and the other type of study includes those that addressed the potential molecular mechanisms underlying DM carcinogenesis. Within the latter group, some investigated the length of the pathogenic repeat expansion in the tumoral tissue of patients with DM1, and a few of those demonstrated that a longer CTG expansion was present in tumoral cells when compared with non-tumoral cells from the same individual [25,26,27,28].

Table 2 summarizes the main findings from the studies that reported an increased risk of specific cancer types in the DM (mostly DM1) population [19,20,22,23,29,30,31,32]. We recently reported a differential cancer risk in our DM cohorts by using for the first time a non-DM muscular dystrophy group as a comparison group. Thus, we identified an increased risk of sex-related and non-sex-related cancers in patients with DM1 and patients with DM2, respectively. The cause of this different type of cancer risk between DM1 and DM2 is unknown. Furthermore, we observed an overall increased cancer risk in DM versus the non-DM muscular dystrophy cohort, which supports the suspicion that this increased cancer risk is intrinsically related to DM rather than a muscular dystrophy diagnosis [33]. However, two limitations that most of these studies share are their retrospective design and difficulties in controlling or adjusting for well-known cancer risk factors, such as sun exposure or smoking. Bianchi et al. reported an increased risk of tumors in DM1 that was not related to lifestyle habits, which reinforces the notion that DM1 genetic defects directly contribute to the increased cancer risk in these patients [34]. 

The second group of studies focuses on the potential molecular mechanisms underlying DM carcinogenesis, although the definitive pathophysiology is still uncertain (Table 3). Several mechanisms such as the upregulation of the *Wnt/B*-catenin pathway and/or alterations of the mRNA transcripts encoding tumor suppressor genes or oncogenes have been postulated [35,36]. Thus, the presence of somatic mutations within the gene encoding for *B*-catenin (*CTNNB1*) and the nuclear accumulation of this protein as a result have been observed in DM tumoral tissue, and these molecular observations have been associated with cell proliferation and differentiation processes that predispose to tumorigenesis [37,38,39]. In addition to *CTNNB1* mutations, some have hypothesized the upregulation of *B*-catenin via the *Wnt* signaling pathway or the overexpression of a second gene (*PLAG1*); the latter has been shown in a mouse model of pleomorphic adenomas of the salivary glands [35,40]. Furthermore, *DMPK* has been reported to be a member of a subfamily of tumor suppressor genes, and interestingly, CTG repeat abnormalities within this gene have also been identified in the tumoral tissue of non-DM patients, prompting one to consider a role of this specific gene in carcinogenesis that is not related to DM muscular dystrophy phenotype [41,42]. Whether the number of pathogenic expanded repeats positively correlates with cancer or tumoral risk in DM is uncertain. Of note, pathogenic CCTG repeat expansions in DM2 are often very large (and technically difficult to determine). One would expect that patients with DM2 have a higher cancer risk than patients with DM1 if longer expansions were unequivocally associated with cancer occurrence. However, carcinogenesis in DM2 has been much less studied than that in DM1, and, as mentioned above, DM2 is frequently underdiagnosed, possibly because it is associated with more variable and less severe skeletal muscle manifestations than DM1. So, it is still plausible that cancer estimates in DM2 are higher than currently recognized. Interestingly, *CNBP* has been reported to be a key transcriptional regulator of tumor-promoting genes to control tumor cell biology [43]. On the other hand, in ours and others’ experience, no correlation between the length of CTG repeats and cancer was observed in patients with DM1 [33,44]. However, it has been reported that the length of the unstable CTG repeat expansion increases over a patient’s lifetime [45]; this observation and the well-known somatic mosaicism of this muscular dystrophy prompt one to consider aging as a stronger risk factor for cancer in patients with DM than in the general population.

In summary, each of the aforementioned mechanisms on their own do not appear to fully account for DM carcinogenesis. It is possible that more than one mechanism is necessary and that carcinogenesis in DM1 differs from that in DM2. Future investigations are needed to elucidate the reason for the higher cancer risk in patients with DM, which may also provide new insights into the carcinogenesis process in general. 

## 4. Pilomatricomas

The most typical and almost pathognomonic tumor described in the DM1 population is the pilomatricoma (or pilomatrixoma, or calcifying epithelioma of Malherbe). This is a benign calcifying skin tumor derived from hair matrix cells, which usually presents as a firm nodule on the scalp, face, or neck area. Pilomatricomas are relatively rare tumors in the adult population, and when present, a diagnosis of DM1 should be considered. However, these benign tumors are the second most frequent skin tumor in childhood, and a female preponderance and the tendency to present as a single lesion within this pediatric group have been reported [46]. They rarely become malignant or recur after surgical excision, which is the curative treatment [47]. 

The presence of multiple pilomatricomas should prompt the clinician to suspect an underlying syndrome or disorder, such as Turner syndrome, Rubinstein–Taybi syndrome, familial adenomatous polyposis-related syndromes, or DM1 [48,49,50]. Around ~10% of patients with DM1 may have pilomatricomas, and in this patient subgroup, these tumors appear to be more frequent in males.

Histologically, the tumor appears as an unencapsulated multinodular lesion within the dermal compartment that extends focally into the subcutis [51]. Mutations of the catenin-β gene (*CTNNB1*), a downstream effector of the *wnt* signaling pathway that has been postulated to be involved in DM carcinogenesis, have been identified in the pilomatricomas of patients with DM1 [38,52]. Thus, *CTNNB1* may have oncogenic properties, and although these benign tumors rarely become malignant, current consensus-based care recommendations advise patients to be referred to surgeons for safe excision [53]. 

## 5. Other Benign Tumors

While the association between pilomatricomas and DM1 (DM formerly) has been well known since 1965 [54], little is known about the DM risk of other benign tumors. In their review of the literature, Mueller et al. summarized a total of 39 articles reporting a variety of benign tumors (excluding pilomatricomas) in 47 patients with DM [35]. Their literature review suggests that thymomas, pituitary adenomas, pleomorphic adenomas of the parotid gland, parathyroid adenomas, insulinomas, colonic adenomas, and benign skin tumors are frequent types of benign tumors in patients with DM. Table 4 summarizes the studies that focused on benign tumors in the DM population [24,55,56]. Alsaggaf et al. did not find significant differences in the age of the first benign tumor diagnosis between patients with DM1 and the general population; however, they observed that specific tumors (such as colorectal polyps, nervous system benign tumors, and thyroid nodules) were more frequently identified at an earlier age in patients with DM1 than in the general population [56]. 

It is also important to emphasize that patients with DM tend to develop multiple (or more than one) benign tumors or cancers in their lifespan, either affecting the same or different organs. Thus, multiple pilomatricomas (which may be the presenting sign of DM), basal cell carcinomas, thymomas, intestinal carcinoid tumors, and multiple endocrine adenomatosis type 2 have been reported [57,58,59,60,61,62,63]. 

## 6. Current Malignancy Screening Recommendations in DM

The consensus-based care recommendations for adults with DM1 (2018) regarding tumor detection advise the following: (1) look for pilomatricomas (hair-matrix-cell-derived benign calcifying skin tumors) and refer to surgeons for safe removal; (2) train patients to detect pilomatricomas; (3) follow general population cancer screening guidelines, particularly for breast, testicular, cervical, and colon cancers; (4) evaluate suspicious new central nervous system, abdominopelvic, and thyroid symptoms for possible cancer; and (5) see full recommendations at myotonic.org/clinical-resources. Similar recommendations were published for adult patients with DM2 (2019), except for the detection of pilomatricomas, which are associated with DM1 only [53,64]. 

A few considerations should be taken into account: (a) cancer screening recommendations for the general population vary across countries, so an adult patient with DM is currently subjected to the recommended malignancy screening in their home country; (b) DM-increased cancer risk is thought to be due to the genetic defect that these patients carry, so their risk of specific cancers may be increased at an earlier age than that of the general population, and malignancy screening may be needed sooner; and (c) adherence to current malignancy screenings may be impaired in this patient population, and, thus, reduced mobility and cognitive difficulties may prevent them from having the recommended testing to detect cancer at an early stage. Furthermore, cancer management often requires general anesthesia, sedatives, and opiates, which patients with DM are well-known to have an increased susceptibility to. 

Thus, we believe that there is a need to investigate whether patients with DM would benefit from more specific malignancy screening guidelines than currently recommended, and, in the meantime, clinicians should emphasize the importance of being adherent to the current cancer screening evaluations recommended for the general population within the multidisciplinary care that these patients should receive.

## 7. Susceptibility to General Anesthesia, Opiates, and Sedatives

Patients with cancer often need to undergo procedures or surgeries for diagnostic or therapeutic purposes that require the administration of general anesthesia, opiates, or sedatives. For this reason, we considered it useful for the purpose of this review to comment on the precautions that medical teams need to be aware of when performing any intervention that requires general anesthesia, sedation, or intensive pain control in these patients. Patients with DM are at increased risk of perioperative complications, which are partly due to their known susceptibility to these agents. 

The earliest description of the risks associated with a myotonic disorder dates back to 1915 when a patient with myotonia congenita (Thomsen’s disease) became cyanotic and unresponsive after the systemic administration of ether [65]. A few more case reports were reported in the following years [66,67]. In addition, a retrospective study of 219 patients with DM showing a perioperative complication rate of 8.2% mostly related to respiratory events (atelectasis, pneumonia, and acute ventilatory failure) was published [68]. In 1960, surgical and anesthesia management to minimize the risks of complications in patients with DM and other myotonic disorders became a priority for many specialists [69]. In addition to this increased sensitivity to anesthetic medications and opioids, cardiac involvement, a restrictive ventilatory pattern, and muscle weakness were also identified as aggravating risk factors for the occurrence of DM perioperative complications. Thus, the muscular impairment rating scale (MIRS), which is a measure of muscle disease severity, has been proposed as a tool to predict a higher operative risk in both adults and children with DM [70,71]. 

In addition to predictive tools and general awareness, communication and the coordination of care among different specialists (neuromuscular specialists, primary care physicians, surgeons, anesthesiologists, pulmonologists, and cardiologists) are of paramount importance. 

In the pre-operative phase, patients with DM should undergo a thorough evaluation with both a primary care provider and a neuromuscular specialist, who will direct the patient to other specialists (cardiology and pulmonology) if clinically indicated. An evaluation by anesthesiologists at this stage is also highly recommended to assess for potential difficulties with airway access and the effectiveness of coughing and to gather prior history of hypoxia, sleep apnea, or coexisting cardiac pathologies that may impact anesthesia management. At the time of surgery, in addition to the standard intraoperative monitoring techniques, patients with DM might require other measures, such as the placement of an arterial line, an external pacemaker, or even invasive cardiac monitoring devices (catheters or central lines). Hypothermia and shivering can induce myotonic contractures; therefore, caution must be taken in keeping the operating room warm, as well as closely monitoring the patient’s body temperature [72]. Regarding the selection of anesthetic medications, patients with DM are generally more sensitive to etomidate, thiopental, and propofol, and lower doses of these agents with a slow titration (if necessary) should be considered [73]. In terms of muscle relaxants, the response to succinylcholine in patients with DM is unpredictable, and this agent must be avoided, as it tends to induce exaggerated muscle contractures, spasms of masseter muscles, or even laryngospasms, all making intubation very challenging, and therefore increasing the risk of complications or even death [74]. When a muscle relaxant is needed, a non-depolarizing agent with a short recovery index, such as rocuronium or cis-atracurium, should be chosen [75]. Finally, patients with DM often wake up slowly from anesthesia and may require a prolonged period of time to be weaned off from ventilatory support. For this reason, measures such as an awake extubation and/or post-procedural monitoring in an intensive care unit need to be considered [76]. Adverse reactions following general anesthesia and sedation appear to be more frequent in DM1 than DM2, but surgeries tend to also be associated with a decline in baseline function in the latter group. Thus, a retrospective study that included 121 adult patients with DM2 reported worsening symptoms of their underlying muscular dystrophy, including myalgias, cramps, and a decline in muscle strength in 14.9% of patients following general anesthesia [77]. 

The European Neuromuscular Centre recently released a consensus statement on anesthesia measures in patients with neuromuscular disorders [78]. Similarly, consensus-based care recommendations for adults with DM advise one to follow Myotonic Dystrophy Foundation’s Practical Suggestions for the Anesthetic Management of a patient with DM (full recommendations at myotonic.org/clinical-resources) before any surgery or procedure requiring anesthesia [53,64]. Although adverse reactions following general anesthesia and sedation appear to be more frequent in DM1 than DM2, the same recommended guidelines apply to the latter patient subgroup. Some of the recommendations include close respiratory and cardiac monitoring and the avoidance of early discharge home following the procedure or surgery, the administration of antacids to minimize aspiration, and the avoidance of any myotonia triggers (such as cold, pain, or any medication that increases muscle membrane excitability) and opiates. Table 5 summarizes the anesthesia guidelines that are recommended in this patient population.

## 8. Conclusions and Future Directions

Patients with DM1 and patients with DM2 have an increased risk of both benign tumors and malignancies compared to the general population. The exact mechanism that accounts for such risk is yet to be defined, but the nuclear accumulation of the β-catenin protein, the activation of the *Wtn* signaling pathway, and the upregulation of oncogenic genes have been postulated [35,36,37,38,39,40,41,42,43]. Of note, GSK3Β is an intracellular regulator kinase that has been demonstrated to be dysregulated in multiple DM1 tissues and considered to be an attractive therapeutic target in this muscular dystrophy. Thus, AMO-02 (Tideglusib) inhibits the activity of this kinase, and it has been shown to correct molecular and behavioral abnormalities in preclinical models and to be safe in a recent Phase II clinical trial in young patients with DM1 [79]. Whether this therapeutic approach lowers cancer risk in DM is unknown and might be a subject of further investigations in future clinical trials or post-marketing studies. In the meantime, prospective studies are needed to investigate the incidence of each specific cancer in the DM population in order to develop more stringent screening guidelines tailored to such incidence. Currently, in addition to close heart and breathing evaluations, monitoring the adherence of patients with DM to the recommended malignancy screening guidelines for the general population and having a low threshold to suspect tumorigenesis in patients with DM are highly recommended. Since patients with DM may develop more than one cancer, it is prudent to intensify malignancy screenings in patients with DM with a prior cancer history. 

Moreover, caution needs to be taken for patients with DM undergoing surgical procedures, as they have an increased susceptibility to anesthesia, opiates, and sedatives. A multidisciplinary team (which includes primary care physicians, neurologists, pulmonologists, cardiologists, anesthesiologists, and surgeons) is recommended to coordinate pre-, intra-, and post-operative care and to avoid anesthesia-related complications. Lastly, the medical team should plan for a longer than usual inpatient stay after any procedure requiring general anesthesia in patients with DM to account for the possibility of prolonged drug effects causing life-threatening complications.

## Figures and Tables

**Figure 1 jcm-12-01939-f001:**
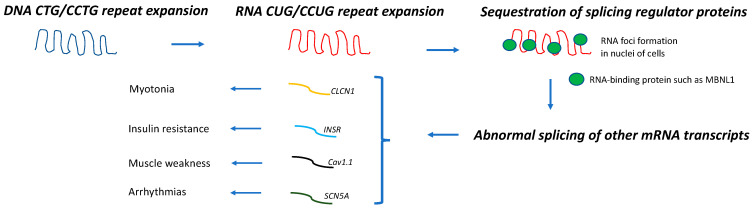
Pathogenic mechanism in DM. DNA repeat expansions are transcribed into mRNA molecules that accumulate in nuclei of cells and sequester splicing regulator proteins, such as MBNL1. As a result, abnormal splicing of a subset of other mRNA occurs, leading to different disease manifestations [6,7,8,9,10].

**Table 1 jcm-12-01939-t001:** Cancer-related death in DM.

Study (Ref)	Country/Registry	Unspecified DM (n)	DM1 (n)	Age at Death (Years Old)	Cancer-Related Death (% of Patients)
de Die-Smulders CE et al., 1998 [14]	The Netherlands	180	---	Mean: 54 95% IC: 52.0–56.7	10%
Mathieu J et al., 1999 [15]	Canada	367	---	Mean: 53.2 (range: 24–81)	11%
Mladenovic J et al., 2006 [16]	Belgrade	---	101	Mean: 56.7 Average mortality rate: 0.5/10^6^	1%
Groh WJ et al., 2011 [17]	United States	---	406	Mean: 54 (range: 21–79)	~6%
Gadalla SM et al., 2013 [18]	Swedish Patient Registry/ Swedish Cause of Death Registry	1081	---	Median: 49.8 95% CI: 39.8–53.8	10%
Fernandez-Torron et al., 2016 [19]	Spain	---	424	---	15.3%

**Table 2 jcm-12-01939-t002:** More frequent types of cancers in DM1 cohorts.

Reference	N	Thyroid	Cutaneous Melanoma	Pancreas	Colon	Endometrium	Ovary	Prostate	Testes	Brain	Eye
Gadalla et al., 2011 [20]	1658	↑SIR 7.1 (1.8–19.3)	–––	↑SIR 3.2 (1.0–7.6)	↑SIR 2.9 (1.5–5.1)	↑SIR 7.6 (4.0–13.2)	↑SIR 5.2 (2.3–10.2)	SIR 0.7 (0.2–1.9)	SIR 1.4 (0.1–6.8)	↑SIR 5.3 (2.3–10.4)	↑SIR 12.0 (2.0–39.6)
Win et al., 2012 [22]	307	↑SIR 5.54 (1.80–12.93)	SIR 2.05 (0.42–6.00)	–––	SIR 1.09 (0.22–3.18)	SIR 1.07 (0.03–5.98)	SIR 1.66 (0.04–9.25)	SIR 2.21 (0.95–4.35)	SIR 5.09 (0.62–18.38)	SIR 1.54 (0.04–8.57)	↑SIR 27.54 (3.34–99.49)
Mohamed et al., 2013 [29]	109	–––	RR 7.1 (0.8–25.8)	–––	RR 5.0 (0.6–18.2)	↑RR 21.7 (2.4–78.5)	RR 9.3 (0.1–51.5)	–––	–––	–––	–––
Abbott et al., 2016 [23]	281	RR 3.78 (0.67–13.65)	RR 0.89 (0.0–4.20)	–––	RR 2.15 (0.11–11.99)	↑RR 6.98 (1.24–25.22)		RR 1.43 (0.25–5.16)	↑RR 10.74 (1.91–38.79)	–––	–––
Fernandez-Torron et al., 2016 [19]	424	↑SIR 23.33 (9.38–48.08)	SIR 1.72 (0.04–9.61)	–––	SIR 2.06 (0.94–3.92)	↑SIR 6.86 (2.23–16.02)	↑SIR 8.33 (1.72–24.31)	SIR 0.46 (0.06–1.67)	SIR 14.25 (0.35–79.6)	↑SIR 9.80 (3.18–22.88)	
Wang et al., 2018 [30]	1061	–––	HR: 2.40 (0.56–10.31)	–––	–––	–––	–––	–––	–––	–––	–––
Alsaggaf et al., 2018 [31]	927	↑HR 15.93 (2.45–103.64)	↑HR 5.98 (1.24–28.79)	↑HR 2.96 (0.30–29.38)	HR 1.82 (0.32–10.31)	↑HR 14.88 (2.14–103.67)		–––	–––	HR 4.99 (0.46–53.78)	–––
				HR 1.81 (0.21–15.23)	HR 1.12 (0.37–3.45)	HR 0.32 (0.04–2.62)				HR 2.03 (0.23–17.68)	
Emparanza et al., 2018 [32]	2779 (meta-analysis)	↑pSIR = 8.52 (3.62–20.1)	↑pSIR = 2.45 (1.31–4.58)	–––	↑pSIR = 2.2 (1.39–3.49)	↑pSIR = 7.48 (4.72–11.8)	↑pSIR = 5.56 (2.99–10.3)	–––	↑pSIR = 5.95 (2.34–15.1)	–––	–––

SIR: Standardized International Ratio; pSIR: pooled SIR; RR: Relative Risk; HR: Hazard Ratio; (): 95% IC.

**Table 3 jcm-12-01939-t003:** Postulated pathogenic mechanisms underlying DM carcinogenesis.

Pathogenic Mechanism	References
1. Upregulation of Wnt/B-catenin pathway	[35]
2. Alterations of mRNAs transcripts (tumor suppressor genes or oncogenes)	[36]
3. B-catenin mutations (*CTNNB1*)	[37,38,39]
4. *PLAG1* overexpression	[40]
5. *DMPK* repeat expansion	[41,42]
6. *CNBP* repeat expansion	[43]

**Table 4 jcm-12-01939-t004:** Frequency of benign tumors reported in the literature.

Ref.	Type of Study/Country	Patients with DM (N)	N of Tumors (N)	Thyroid	Skin	GI	Breast	Sex (Female)-Related	Sex (Male)-Related	Hematologic	Salivary Glands	Brain	Other
Ben Hamou et al., 2019 [55]	Retrospective Observational/ US	115 (DM1)	N/A	Nodule = 61% (70/115), of which 50% with benign cytology *									
Alsaggaf et al., 2018 [24]	Cross-Sectional, Self-reported/ UK	220 (DM1 and DM2)	39 (N/A)	6 (15.4) ^	4 (10.3%)	0 (0)	2 (5.1)	11 (28.2)	0 (0)	0 (0)	4 (10.3)		10 (25.6)
Alsaggaf et al., 2018 [31]	Retrospective Observational/ UK	927 (DM1)	138 (14%) vs. 844 (6%) DM1 vs. DM1-free	HR 10.4 (3.9–27.5)	HR 1.4 (1.1–1.9)	HR 4.3 (1.8–10.4)		Uterine fibroids: HR: 2.7 (1.2–5.9); Uterine polyps: HR 9.6 (1.2–77.5)				HR 8.4 (2.5–28.5)	
Mueller et al., 2009 [35]	Review of Case Reports/ US	N/A	N/A	^^	^^	^^		^^	^^	^^	^^		^^

Legend: N = number of tumors not patients. HR = comparing between DM and control group, with CI reported only when significant. *: 22.7% non-diagnostic; 27.2% indeterminate. ^: referred to as endocrine in general. ^^: reported tumors in the individual study, but HR or other risk factor data not available.

**Table 5 jcm-12-01939-t005:** Considerations regarding anesthesia in DM1 and DM2.

**Pre-operative assessment**	History: age at diagnosis, cardiac, pulmonary problems, obstructive sleep apnea, central apnea, prior history of surgery-related complications, medications.
	Exam: general and neurological examination, muscle function (MIRS), craniofacial abnormalities (indicating difficult airway).
	Evaluations: blood glucose, electrolytes, hemoglobin, LFTs, thyroid function, CK, ECG, chest X-ray, echocardiogram, Holter examination, PFTs, sleep oximetry or polysomnography.
	Specialists: primary care provider, neuromuscular specialist, cardiologist, pulmonologist, anesthesiologist.
**Operative management**	Schedules: Patients with DM should avoid prolonged fasting and hypoglycemia.
	Monitor: pulse oximetry, telemetry, invasive cardiac monitoring.
	Temperature control: both that of the patient and that of the operating room.
**Induction agents**	Premedication: avoid. Benzodiazepines can cause central respiratory depression.
	Halogenated agents: no increased risk of MH.
	Hypnotics: slow titration. Propofol-induced pain can provoke myotonia. Etomidate-induced pain can provoke myoclonic movements.
	Muscle relaxants: unpredictable response to succinylcholine. Non-repolarizing agents, such as rocuronium or cis-atracurium, are preferred.
	Opioids: fentanyl, sufentanil, or remifentanil have been used.
	Reversal of muscle relaxation: cholinesterase inhibitors are contraindicated.
**Recovery**	Extubation: fully awake and with the support of NIV.
	Prolonged PACU stay, with continuous monitoring of O2 and surveillance for signs of rhabdomyolysis. Risk of delayed respiratory problems, aspiration, and ileus.
	Analgesia: NSAIDs and paracetamol are safe. Caution with opioids. Tramadol has been used.

NIV: non-invasive ventilation, PACU: pediatric acute care unit, NSAIDs: non-steroid anti-inflammatory drugs, CK: creatine kinase, LFTs: liver function tests, PFTs: pulmonary function tests, MH: malignant hyperthermia, EKG: electrocardiogram, MIRS: Muscular Impairment Rating Scale.

## Data Availability

Not applicable.

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
