# Peer review of "Cancer and Myotonic Dystrophy"

_jcm, 2023, doi:10.3390/jcm12051939_

Round 1
Reviewer 1 Report
The manuscript by D’Ambrosio et al. reviewed literature data concerning cancer risk in patients affected by Myotonic Dystrophy, the most common muscular dystrophy in adults. Molecular mechanisms responsible for carcinogenesis in DM are described and the main studies investigating cancer types and risk are summarized and discussed. On these bases, authors propose some evaluations to be considered as malignancy screening in DM patients and discuss DM susceptibility to general anesthesia and sedatives, generally used in the management of cancer. The review underlines the need of monitoring DM patients’ adherence to malignancy screenings and the importance to design studies that determine whether they would benefit from a more intensified cancer screening than general population.
Although not innovative, the topics of the review is interesting because it combines molecular and clinical studies focused on carcinogenesis in DM patients. However, some points need to be improved and addressed before the review is considered suitable for publication.
Major points:
1) References on page 2 and 3, sections 1 and 2, seems to be incomplete. The ones cited do not fully support what is written. It is also advisable to cite the reference for the example shown at page 3 lanes 84-85.
2) The authors should specify how they conducted the literature investigation (search and exclusion criteria).
3) During the evaluation of the bibliography, I was unable to consult reference 33, by the same authors, which is crucial for the development of the discussion and for supporting the claims made.
4) Cited bibliography is incomplete. Please discuss the following paper reporting a clinical study on cancer risk in DM:
Erratum to: Increased risk of tumor in DM1 is not related to exposure to common lifestyle risk factors. Bianchi ML, Leoncini E, Masciullo M, Modoni A, Gadalla SM, Massa R, Botta A, Rastelli E, Terracciano C, Antonini G, Bucci E, Petrucci A, Costanzi S, Santoro M, Boccia S, Silvestri G.J Neurol. 2016 Mar;263(3):499. doi: 10.1007/s00415-016-8068-5.PMID: 26914929.
5) Please discuss the role of the CNBP gene and protein in tumorigenesis and the possible implication for cancer risk in DM2 patients:
Lee E, Lee TA, Yoo HJ, Lee S, Park B. CNBP controls tumor cell biology by regulating tumor-promoting gene expression. Mol Carcinog. 2019 Aug;58(8):1492-1501. doi: 10.1002/mc.23030. Epub 2019 May 13. PMID: 31087358.
6) The discussion of the molecular mechanisms at the basis of carcinogenesis in DM would benefit from a Table or a Figure recapitulating the state of the art and the main pathogenic hypotheses. This latter part should be removed from Figure 1 and illustrated in a dedicated Table or Figure related to “Carcinogenesis and DM” section.
Minor revisions:
1) OMIM entry must be added when referring to disease genes and genetic diseases
2) According to the official nomenclature, genes and corresponding mRNA or transcript must be italicized
3) Please add the number of Figure at page 2 lane 52 and at page 4 lane 132
4) Correct the term mIRS into MIRS at pag 8 lane 257
5) Check sentence at pag. 3 lanes 109-111
Author Response
Reviewer 1
The manuscript by D’Ambrosio et al. reviewed literature data concerning cancer risk in patients affected by Myotonic Dystrophy, the most common muscular dystrophy in adults. Molecular mechanisms responsible for carcinogenesis in DM are described and the main studies investigating cancer types and risk are summarized and discussed. On these bases, authors propose some evaluations to be considered as malignancy screening in DM patients and discuss DM susceptibility to general anesthesia and sedatives, generally used in the management of cancer. The review underlines the need of monitoring DM patients’ adherence to malignancy screenings and the importance to design studies that determine whether they would benefit from a more intensified cancer screening than general population.
Although not innovative, the topics of the review is interesting because it combines molecular and clinical studies focused on carcinogenesis in DM patients. However, some points need to be improved and addressed before the review is considered suitable for publication.
Major points:
1) References on page 2 and 3, sections 1 and 2, seems to be incomplete. The ones cited do not fully support what is written. It is also advisable to cite the reference for the example shown at page 3 lanes 84-85.
Response: We have added Tanboon J Curr Opin Neurol 2022 as reference 13 as citation for the association between TIF1 antibody and cancer-associated dermatomyositis. We have reviewed all references on page 2 and 3 and made sure they are completed citations and that are cited properly.
2) The authors should specify how they conducted the literature investigation (search and exclusion criteria).
Response: We thank Reviewer for this important point. We have added under “Mortality in DM” section the following: “We conducted a review of literature using “myotonic dystrophy [Title/Abstract], cancer [Title/Abstract], mortality, and death [Title/Abstract] in Pubmed to identify research articles that studied death due to cancer in DM.”
We have also added the following under “Carcinogenesis in DM” section: “We conducted a literature search in PubMed using “myotonic dystrophy [Title/Abstract] and cancer {Title/Abstract] to identify those research articles that investigated frequency, type, and pathogenic mechanism of cancer in DM. Research articles that did not aim at investigating these aspects and non-research articles were also reviewed and referenced here only if considered clinically relevant for the purpose of this review.”
3) During the evaluation of the bibliography, I was unable to consult reference 33, by the same authors, which is crucial for the development of the discussion and for supporting the claims made.
Response: We have completed the info (ref. 33); our manuscript has been very recently published.
D’Ambrosio ES, Chuang K, David WS, Amato AA, Gonzalez-Perez P. Frequency and Type of Cancers in Myotonic Dystrophy. A Retrospective Cross-Sectional Study. Muscle and Nerve 2023 (in press). PMID: 36790141
4) Cited bibliography is incomplete. Please discuss the following paper reporting a clinical study on cancer risk in DM:
Erratum to: Increased risk of tumor in DM1 is not related to exposure to common lifestyle risk factors. Bianchi ML, Leoncini E, Masciullo M, Modoni A, Gadalla SM, Massa R, Botta A, Rastelli E, Terracciano C, Antonini G, Bucci E, Petrucci A, Costanzi S, Santoro M, Boccia S, Silvestri G.J Neurol. 2016 Mar;263(3):499. doi: 10.1007/s00415-016-8068-5.PMID: 26914929.
Response: We thank the Reviewer for this important reference. We have added the following under “Carcinogenesis in DM” section:“Bianchi et al. reported an increased risk of tumors in DM1 that was not related to life-style habits which reinforces a direct contribution of DM1 genetic defect to cancer risk [34]”
5) Please discuss the role of the CNBP gene and protein in tumorigenesis and the possible implication for cancer risk in DM2 patients:
Lee E, Lee TA, Yoo HJ, Lee S, Park B. CNBP controls tumor cell biology by regulating tumor-promoting gene expression. Mol Carcinog. 2019 Aug;58(8):1492-1501. doi: 10.1002/mc.23030. Epub 2019 May 13. PMID: 31087358.
Response: We have added the following under “Carcinogenesis in DM” section: “Interestingly, CNBP has been reported to be a key transcriptional regulator of tumor-promoting genes to control tumor cell biology [43].”
6) The discussion of the molecular mechanisms at the basis of carcinogenesis in DM would benefit from a Table or a Figure recapitulating the state of the art and the main pathogenic hypotheses. This latter part should be removed from Figure 1 and illustrated in a dedicated Table or Figure related to “Carcinogenesis and DM” section.
Response: As Reviewer suggested, we have modified Figure 1 removing cancer pathogenesis in DM and added a Table summarizing main pathogenic hypotheses in DM carcinogenesis. This new table is now Table 3.
Minor revisions:
1) OMIM entry must be added when referring to disease genes and genetic diseases
Response: OMIM 160900 for DM1 and 602668 for DM2 have been included within first paragraph of Introduction.
2) According to the official nomenclature, genes and corresponding mRNA or transcript must be italicized
Response: All genes and mRNAs have been italicized.
3) Please add the number of Figure at page 2 lane 52 and at page 4 lane 132
Response: Figure 1 has been added.
4) Correct the term mIRS into MIRS at pag 8 lane 257
Response: This term has been corrected.
5) Check sentence at pag. 3 lanes 109-111
Response: We have reviewed the statement that corresponds to original lanes 109-111 and modified as follows: “adherence to current malignancy screenings may be impaired in this patient population, thus; reduced mobility and cognitive difficulties may prevent them from having the recommended testing to detect cancer at early stage.”
Reviewer 2 Report
The review by D´Ambrosio and González-Pérez authors entitled “Cancer and Myotonic Dystrophy” presents an up-to-date recapitulation of cancer manifestations in myotonic dystrophy rare diseases. This is one of the most intriguing downstream clinical manifestations of these rare diseases.
The review contains a very complete list of publications being aware of this connection, perceiving with expectancy every day a clearer connection confirmation. The manuscript also introduces novel subdivisions, very useful for readers, to explain differences between malign and benign tumors present in DM, or explain deeper some of the most frequent.
Still, the manuscript has space for the improvement through this minor issues:
§ A literature search displays that key factors involved in DM pathogenesis, such as Muscleblind, Celf1, and even others not mentioned where therapeutic developments are ongoing, like GSK3β (Tideglusib), also have a robust connection with cancer regulation. This information should be somehow in this review, at least as a “discussion point” or through section 4 where mechanistic evidences displayed.
§ The introduction for DM pathologies is very “typical”. This reviewer is being reading this type of text since almost twenty years ago. The intensive last years of research on this area has provided new routes/mechanistic (for example involving muscleblind repression by specific miRNAs). It is not a matter of making the introduction larger but of defining more precisely the current situation.
§ The manuscript does not seem to follow a logical order. For example, the section 2 describing screening recommendations would be better to read it later in through the manuscript, once all the evidences and numbers provided for the connection.
§ Hypothesis for sex-related cancers in DM1-DM2? It is said through the manuscript as occurring but no potential explanation identified
§ Conclusion section has format issues, and also needs of text improvement for example including mechanistic hypothesis or the proposal of specific improvements in the management of the patients related to cancer manifestations/diagnosis.
§ The section 7 “…anesthesia…” deviate the attention from the cancer appearance in DM. It is clear that cancer interventions will need of patients´ management but the text rapidly moves to a more general situation of anesthesia intervention. Reduce it? Connect better with the total number or % of cancer interventions in DM patients? Is this number of cancer related interventions significant relative to other type of interventions that need anesthesia?
Other minor concerns
§ Figure 1 has also space for improvement on its information and graphics
§ Lines 36-37 could be improved on the information are giving. I would recommend putting the information between commas at the end and informing about potential “founder effects” on these populations where the prevalence is much higher
§ Figure 1 legend needs of references for the data mentioned
§ Section “Myopathies and cancer” lacks of the numbering. Should be the 2, and from here the following also will change in their number. Also all the sections are in italic font, except the 1. That is in bold and non-italic
§ Lines 69-79: a lot of information for different myopathies and cancer, connected to only one reference (number 11) that looking the title seems not to be related to “muscle”
§ Lines 87-88: authors make a general assumption of DM and cancer connection but without references
§ Line 123: what is a “regular” % of cancer deaths in whole population. This number would be useful to compare with what is happening in DM
§ Line 132: the (Figure) lacks the 1
§ Table 2 title: remove “predominantly” since it is not clear what this word wants to mean here
§ Include all the abbreviations meaning in the Tables legend. For example, for Table 2 no description for RR. Also, some description of the numbers/ranges displayed is recommended for a better interpretation of Tables. Table 4 does not have legend but several word need of interpretation: NIV, PACU, NSAIDs..
Author Response
Reviewer 2
The review by D´Ambrosio and González-Pérez authors entitled “Cancer and Myotonic Dystrophy” presents an up-to-date recapitulation of cancer manifestations in myotonic dystrophy rare diseases. This is one of the most intriguing downstream clinical manifestations of these rare diseases.
The review contains a very complete list of publications being aware of this connection, perceiving with expectancy every day a clearer connection confirmation. The manuscript also introduces novel subdivisions, very useful for readers, to explain differences between malign and benign tumors present in DM, or explain deeper some of the most frequent.
Still, the manuscript has space for the improvement through these minor issues:
- A literature search displays that key factors involved in DM pathogenesis, such as Muscleblind, Celf1, and even others not mentioned where therapeutic developments are ongoing, like GSK3β (Tideglusib), also have a robust connection with cancer regulation. This information should be somehow in this review, at least as a “discussion point” or through section 4 where mechanistic evidences displayed.
Response: We thank Reviewer for this important point. We have added the following to Conclusions: “Of note, GSK3B is an intracellular regulator kinase that has been demonstrated to be dysregulated in multiple DM1 tissues and considered to be an attractive therapeutic target in this muscular dystrophy. Thus, AMO-02 (Tideglusib) inhibits the activity of this kinase and has been shown to correct molecular and behavioral abnormalities in preclinical models and being safe in a recent Phase II clinical trial in young DM1 patients [79]. Whether this therapeutic approach would lower cancer risk in DM is unknown and might be a subject of further investigations in future clinical trials or post-marketing studies. In the meantime, prospective studies…”
- The introduction for DM pathologies is very “typical”. This reviewer is being reading this type of text since almost twenty years ago. The intensive last years of research on this area has provided new routes/mechanistic (for example involving muscleblind repression by specific miRNAs). It is not a matter of making the introduction larger but of defining more precisely the current situation.
Response: We thank Reviewer for this important pint. We have added relevant recent literature regarding miRNAs role in DM pathogenesis within Introduction. “Furthermore, abnormal regulation of miRNAs in both DM1 tissue and extracellular vesicles in biofluids of DM1 patients have been postulated to play a key role in DM pathogenesis and even serve as disease biomarkers [11].”
- The manuscript does not seem to follow a logical order. For example, the section 2 describing screening recommendations would be better to read it later in through the manuscript, once all the evidences and numbers provided for the connection.
Response: We have modified order, section 2 is now section 6.
- Hypothesis for sex-related cancers in DM1-DM2? It is said through the manuscript as occurring but no potential explanation identified.
Response: We thank Reviewer for this important point. Unfortunately, it is unknown the underlying pathogenic mechanisms that increases risk for sex-related cancer and non-sex-related cancers in DM1 and DM2, respectively, as our recently published study showed [D’Ambrosio et al. Muscle Nerve 2023, ref. 33]. We have added this “The cause of this different type of cancer risk between DM1 and DM2 is unknown”
- Conclusion section has format issues, and also needs of text improvement for example including mechanistic hypothesis or the proposal of specific improvements in the management of the patients related to cancer manifestations/diagnosis.
Response: We have corrected font in this section that we named “Conclusions and Future Directions”. We have included information about Tideglusib phase 2 clinical trial within this section. We have also added the following to last paragraph of the Conclusion section: “Lastly, medical team should plan for a longer than usual inpatient stay after any procedure requiring general anesthesia in DM patients fo the possibility of prolonged drug effect causing life-threatening complications.”
- The section 7 “…anesthesia…” deviate the attention from the cancer appearance in DM. It is clear that cancer interventions will need of patients´ management but the text rapidly moves to a more general situation of anesthesia intervention. Reduce it? Connect better with the total number or % of cancer interventions in DM patients? Is this number of cancer related interventions significant relative to other type of interventions that need anesthesia?
Response: We have reduced this section and connected better with DM-associated cancer which is the subject of this review. We have added: “For this reason, we considered useful for the purpose of this review to comment on the precautions that medical teams need to be aware of when performing any intervention that requires general anesthesia, sedation, or intensive pain control in these patients.”
Other minor concerns
- Figure 1 has also space for improvement on its information and graphics
Response: We have modified Figure 1 as also suggested by Reviewer 1.
- Lines 36-37 could be improved on the information are giving. I would recommend putting the information between commas at the end and informing about potential “founder effects” on these populations where the prevalence is much higher.
Response: We have modified these sentence as follows: “The prevalence of DM1 varies widely across different populations being higher in northern Sweden, Quebec, and Basque region suggesting a founder effect (prevalence=~1/4,000-8,000) [1,2].”
- Figure 1 legend needs of references for the data mentioned
Response: We have included reference in Figure 1 legend.
- Section “Myopathies and cancer” lacks of the numbering. Should be the 2, and from here the following also will change in their number. Also all the sections are in italic font, except the 1. That is in bold and non-italic
Response: We have considered “Myopathies and cancer” a subsection of Introduction. We have changed to non-Italic the titles of the main sections of this Review.
- Lines 69-79: a lot of information for different myopathies and cancer, connected to only one reference (number 11) that looking the title seems not to be related to “muscle”.
Response: The reference is a chapter of a book written by one of the authors (Dr. Gonzalez Perez). We have included the title of chapter (Myopathies and Cancer) within the reference that is now 12.
- Lines 87-88: authors make a general assumption of DM and cancer connection but without references
Response: We have modified these lanes as follows since we have included all references later in the text: “ On the other hand, for myotonic dystrophies (for which we no cure exists at this time) there are not stratifiers of cancer risk to guide malignancy screening in these patients.”
- Line 123: what is a “regular” % of cancer deaths in whole population. This number would be useful to compare with what is happening in DM
Response: We would need assistance here. We don’t see the word “regular” on lane 123.
- Line 132: The (Figure) lacks the 1
Response: We have modified Figure as Figure 1.
- Table 2 title: remove “predominantly” since it is not clear what this word wants to mean here
Response: This word has been removed.
- Include all the abbreviations meaning in the Tables legend. For example, for Table 2 no description for RR. Also, some description of the numbers/ranges displayed is recommended for a better interpretation of Tables. Table 4 does not have legend but several word need of interpretation: NIV, PACU, NSAIDs.
Response: We have included meaning of abbreviations below each table including RR: relative risk, NIV: non-invasive ventilation, PACU: Pediatric acute care unit, NSAIDs: non-steroid anti-inflammatory drugs etc.